# Green Approaches on Modification of Xylan Hemicellulose to Enhance the Functional Properties for Food Packaging Materials—A Review

**DOI:** 10.3390/polym15092088

**Published:** 2023-04-27

**Authors:** Petronela Nechita, Mirela Roman (Iana Roman), Silviu Marian Năstac

**Affiliations:** 1Research and Consultancy Center for Agronomy and Environment, Engineering and Agronomy Faculty in Brăila, “Dunărea de Jos” University of Galați, 810017 Braila, Romania; 2Doctoral School of Fundamental and Engineering Sciences, “Dunarea de Jos” University of Galati, 817112 Braila, Romania; 3Research Center for Mechanics of Machines and Technological Equipments, Engineering and Agronomy Faculty in Brăila, “Dunărea de Jos” University of Galați, 810017 Braila, Romania; snastac@ugal.ro; 4Department of Mechanical Engineering, Faculty of Mechanical Engineering, Transilvania University of Brașov, 500014 Brașov, Romania

**Keywords:** hemicelluloses, xylan, acetylation, anhydrides, ionic liquids, enzymes, food packaging, barrier properties

## Abstract

Based on the environmental concerns, the utilisation of hemicelluloses in food packaging has become a sustainable alternative to synthetic polymers and an important method for the efficient utilisation of biomass resources. After cellulose, hemicellulose is a second component of agricultural and forestry biomass that is being taken advantage of given its abundant source, biodegradability, nontoxicity and good biocompatibility. However, due to its special molecular structure and physical and chemical characteristics, the mechanical and barrier properties of hemicellulose films and coatings are not sufficient for food packaging applications and modification for performance enhancement is needed. Even though there are many studies on improving the hydrophobic properties of hemicelluloses, most do not meet environmental requirements and the chemical modification of these biopolymers is still a challenge. The present review examines emerging and green alternatives to acetylation for xylan hemicellulose in order to improve its performance, especially when it is used as biopolymer in paper coatings or films for food packaging. Ionic liquids (ILs) and enzymatic modification are environmentally friendly methods used to obtain xylan derivatives with improved thermal and mechanical properties as well as hydrophobic performances that are very important for food packaging materials. Once these novel and green methodologies of hemicellulose modifications become well understood and with validated results, their production on an industrial scale could be implemented. This paper will extend the area of hemicellulose applications and lead to the implementation of a sustainable alternative to petroleum-based products that will decrease the environmental impact of packaging materials.

## 1. Introduction

Currently, packaging materials are produced from petroleum-based resources using non-degradable polymers, with negative impacts on the environment and sustainability [1,2]. In this context, there is a real need to develop sustainable packaging materials using renewable and natural resources with low impact on the environment, in terms of deterioration, and no competition with food resources. This could be achieved by the efficient exploitation of lignocellulosic biomass resources or agricultural residues [3].

The special attention given to naturally and renewable biopolymers is based on their potential to be used as biodegradable, edible films and coatings for food packaging paper. Compared with conventional petroleum-based synthetic polymers, the packaging materials based on biopolymers from renewable resources offer important environmental benefits as recyclability, biocompatibility including reutilisation and circularity features [4,5,6]. 

Generally, biopolymer films and coatings have good barrier properties for gases or humidity, thus reducing the alteration of food quality and extending the shelf life of foods. In addition, biopolymers act as efficient matrices for embedding different additives in order to confer their active properties to packaging material, such as antimicrobial, antioxidant or nutrient properties. For food packaging and coatings, barrier properties (oxygen, oil, water and water vapours) are the most important aspect for the preservation of food quality and shelf life. These properties are responsible by the control of gas exchange and the prevention of chemical reactions of the food or microbiological contamination and deterioration [7,8]. 

To achieve this, polysaccharides are viewed as the main candidates for substitution with synthetic polymers, because they are non-toxic and biodegradable and have film-forming capacities and provide good barriers to gases, aromas and liquids. However, coatings and films from native polysaccharides present some disadvantages, which are linked to their hydrophilic nature and crystalline structure, i.e., they have low water resistance or are poor barriers against water vapours [9,10,11]. 

After cellulose, hemicelluloses are the second most abundant polysaccharides in nature, representing a promising source of renewable materials to obtain the high-value products [12,13,14,15,16,17]. Xylans are the largest groups of hemicelluloses that are not currently used in food production. They are available in high quantities in wood products (about 30% in hardwood and 10% in softwood) and agriculture residues and are used in agro-industries as by-product from dissolved wood pulp manufacturing [18,19,20]. 

The main limitation of the industrial application of xylan hemicellulose is their high hydrophilic character, resulting from an abundance of free hydroxyl groups distributed along their backbones and side chains (Figure 1) [21].

Based on these features, native hemicelluloses alone are usually unable to form strong and durable films. In a recent study [23], the performance of native hardwood xylan was evaluated regarding the water and oil barrier properties of xylan-coated papers and as protective films for foods packaging. The results revealed a slight improvement of the barrier properties of coated papers and poor film-forming abilities. In its native form, xylan hemicellulose forms high brittle films with reduced mechanical stability. This is a direct consequence of the insufficient chain length of the xylan polymer and its poor solubility [24].

The antimicrobial and antifungal properties of xylan films and paper coated with hardwood xylan hemicelluloses have been reported on in other recent studies [25,26,27,28]. It was concluded that, in its native form, xylan shows moderate antifungal activity and slightly antimicrobial effects against pathogen bacteria (*E.coli*, *S. aureus*).

Nevertheless, the presence of hydroxyl groups in the xylan structure makes it an appropriate candidate for chemical functionalization by attaching hydrophobic groups onto hemicellulose chains using a variety of chemical reactions. In this context, in recent decades, xylan derivatives are gaining importance as the basis of new biopolymeric materials and functional biopolymers. Many methods have been explored for the modification and treatment of hemicelluloses with the aim of improving their performance [29]. As result, a new material with improved properties, such as hydrophobicity, thermal formability and an ability to film form, can be obtained. This could extend the scope of applications for xylan hemicellulose, especially regarding packaging uses, as edible films or as coatings for packaging paper, as well as for biomedical products and drug encapsulation [30,31,32,33].

Interest in the hemicellulose modification is growing rapidly and this is reflected by the increasing number of published and cited papers concerning this topic over the last two decades (Figure 2). This growth is primarily based on the environmental need to replace synthetic polymers and to produce sustainable and biocompatible packaging materials [34,35,36]. 

To broaden hemicellulose applications, a variety of chemical reactions for its functionalization have been reported, such as oxidation, reduction, esterification (i.e., acetylation, propionylation, oleoylation, lauroylation, benzolation or crosslinking) or etherification (i.e., cationization, carboxymethylation or alkoxylation) [37,38,39,40,41,42,43]. 

The aim of this review is to resume and compare the modification methods and approaches for the hydrophobisation and acetylation of hemicelluloses (i.e., xylan) to improve their performances, especially when they are used in paper coatings or films for food packaging. The modification methods with low environmental impacts and their effects on the properties of hemicelluloses will be presented while keeping in mind both the quality performances of obtained derivatives and their practical applications. 

## 2. Modification of Xylan Hemicelluloses by Esterification

### 2.1. Xylan Hemicellulose–Potential Applications and Industrial Availability

Xylan is the major hemicellulose in hardwood and has a wide variety of applications due to its poly-diversity and poly-molecularity. However, until now, the commercial applications of xylan hemicelluloses are limited to xylitol and biofuels obtained through the biological conversion of sugar, starch and vegetable oils [44] and as sources of heat energy [45]. A high quantity of xylan from agricultural residues is burnt or used as animal feedstock. In the packaging industry, xylan is used as an additive for plastics to increase their strength and biodegradability [18,46]. In the past decade, a technology based on xylan hemicelluloses modified with vegetal waxes, as an excellent barrier against water, moisture, grease and oil, was developed and patented by Xylophane AB (Sweden), with their *Skalax^®^* product being available for packaging applications [47]. In addition, xylan is used in the bakery industry to improve the quality of grain flour, and it has a wide therapeutic range, as it is extremely beneficial in the pharmaceutical field as a drug carrier, such as micro-and nanoparticles, as well as film coatings [48]. Based on its biodegradable and biocompatible features, xylan is considered to be a green polymer with important role in the renewability of waste products. In this context, the size of the global xylan market was valued at USD 1.57 billion in 2021 and total xylan revenue is expected to grow by 6% through 2022 to 2029, reaching nearly USD 2.50 billion [49]. The market’s progress must be boosted by research and development focused on improving the industry understanding of xylan biosynthesis and its chemical modifications in order to obtain tailored characteristics to broaden application areas [48]. 

The presence of a large number of active groups (i.e., hydroxyl, carboxyl, and carbonyl) in the structural unit of xylan hemicelluloses facilitates their modification through a series of chemical reactions. As a result, the functional properties of hemicelluloses are improved, making them more appropriate for use in different applications (i.e., food packaging).

Esterification is the most commonly method used for the chemical modification of hemicelluloses, and many studies on acetylated xylan have been published [50,51,52]. The reaction of xylan with propylene oxide and then acetylated, butylated, or allylated hydroxypropyl xylan are other reported approaches [37,53,54]. 

It is known that esterification is a chemical reaction between alcohols and carboxylic acids or one of its derivatives to form an ester. Generally, acid chlorides or acid anhydrides are used as derivatives which react with available hydrophilic OH groups (i.e., C2 and C3 from xylan unit) and replace them with more hydrophobic acetyl groups. This reaction is typically performed in solvents, such as dimethylformamide (DMF), dimethylacetamide/lithium chloride (DMAc/LiCl), dimethylsulfoxide or ionic liquids (ILs), using acid or alkaline catalysis in some cases or in the presence of triethylamine in order to neutralize the generated hydrochloric acid [55]. 

### 2.2. Acetylation of Xylan Hemicelluloses with Long-Chain Anhydrides

Recent approaches used to obtain functionalised xylan with improved hydrophobicity are based on the reaction of xylan with long-chain anhydrides, such as alkyl ketene dimers (AKDs) and alkenyl succinic anhydrides (ASAs) [21]. An additional advantage of this approach is that both anhydrides are well-known as sizing agents for the paper industry and are cheaper and could thus be more easily introduced for commercial application. In this reaction, the hydroxyl groups of xylan hemicelluloses react to form esters or β-keto esters with hydrophobic features. In their research, Hansen and Plackett prepared films using modified birch wood xylan with alkenyl succinic anhydrides (ASAs). The hydrophobicity of obtained films was improved and contact angle was increased from 28° to 65°, even at low degrees of esterification. However, due to the instability of carboxylic groups, the thermal stability of the obtained xylan derivative was lower than that of the original xylan [56].

Recent studies reported the improvement of barrier properties against water, air and oil of packaging papers coated with xylan hemicellulose and functionalised with alkyl ketene dimers [57,58]. In this case, 1.0% of AKDs in xylan dispersion was applied onto the paper’s surface (4.5 g/m^2^), resulting in a reduction of water vapour transmission rate (WVTR) of 35%, compared with untreated paper, and around a three-fold increase in the coated paper’s resistance to air [57]. 

In other research, Tian et al. obtained the prolonging of shelf life for green asparagus, from 5 days to 7 days by coating with a composite film prepared from hemicellulose, nano-cellulose, montmorillonite and alkyl ketene dimers. In this case, the addition of AKDs was able to effectively improve the air barrier properties and decrease the water vapour permeability of the hemicellulose-based film. In addition, the AKDs xylan-modified coating film reduces the secretion of lignification enzymes by adjusting phenolic compounds’ conversion to the lignin monomers. The polymerization of mono-lignin molecules into lignin occurs [59].

The reaction of xylan hemicelluloses with trifluoroacetic anhydride when the fluorinate xylan with enhanced hydrophobicity is obtained, was performed by Grondhal et al. [60]. 

Functionalised xylan with 2-dodecenyl succinic anhydride was used to obtain films for paper lamination. It was observed that the long carbon chains of esterified xylan provide flexibility and hydrophobicity for the obtained films and improve the mechanical and gas barrier properties of laminated papers [12]. These performances are promising for xylan biopolymers to be used in food packaging applications. 

### 2.3. Acetylation of Xylan Hemicelluloses with Acetic Anhydride

Even if there are numerous reported methods of acetylation with different levels of substitution degree and conditions, the utilisation of acetic anhydride as a reagent remains the most basic reaction for achieving acetylated hemicelluloses [61,62]. 

The esterification of xylan hemicelluloses by acetylation is frequently reported to obtain acetylated xylan films with low sensitivity to water, good thermal and mechanical properties, very low oxygen permeability and grease barrier properties [63,64]. Additionally, the acetylated xylan can function as an internal plasticizer by inducing relatively good mechanical properties for a hemicellulose acetate film without the addition of external plasticizer [21,40,65]. The most common reaction for the acetylation of hemicelluloses can be achieved using acetic anhydride or acetyl chloride in formamide/pyridine or other systems [66]. The reaction mechanism is based on the nucleophilic attack of the electron pair of xylan hydroxyl groups through the carbonyl carbon atom of the acetic acid anhydride molecule. As result, the split-off of acetic acid occurs and acetylated carbohydrates are formed. This reaction is often performed through acid or base catalysis (Figure 3). 

Generally, the acetylation of polysaccharides improves the mobility of polymer chains as the effect of acetyl groups prevent the tight-packing of polymer chains and the formation of strong hydrogen bonds. X-ray diffraction confirms this and reveals the acetylated xylan’s amorphous structure [68].

In their thesis, Stepan proved that the films produced from fully acetylated arabinoxylans have similar or superior mechanical and thermal properties compared to highly substituted cellulose acetates (Table 1). In addition, through complete acetylation, the water stability of xylan is improved. Therefore, increasing the water resistance, thermoplasticity and thermostability of xylan biopolymers through full acetylation enables them to be used in a broader field of applications and facilitates a variety of thermo-processing technologies to be applied to the production processes (Table 2). In this context, the acetylated xylan has high potential to be used in the packaging industry as paper coating and wrapping foil for food products [69]. 

Depending on the targeted degree of substitution, different chemical reaction conditions can be used to obtain hemicelluloses acetates. There are also reported results with different substitution degrees obtained from the application of a variety of methods, such as surface esterification in the gas phase, plasma treatment-assisted surface esterification or treatment with glacial acetic acid. Another often used method is the reaction of hemicelluloses with dimethylacetamide (DMAc) using LiCl as catalyst [70,71,72].

Carvalho et al. studied the effect of different degrees of acetylation on the improvement of the thermal stability of xylan isolated from various sources (birch, eucalyptus, spruce, sugarcane bagasse and sugarcane straw), using acetic anhydride and different parameters of reaction. Regardless of molecular weight or xylan type, after acetylation, a substantial improvement in the thermal stability of acetylated xylan samples was obtained. An increase of 17–61 °C and 75–145 °C in the thermal stability of xylan was observed at low and high degrees of acetylation, respectively. Based on these results the acetylation could be an efficient method for increasing of xylan thermal stability and extending the area for the utilisation of xylan derivatives as thermoplastic or packaging materials [73].

The purity of xylan sample has a high influence on the acetylation process. For example, samples with low level of impurities could reach higher degree of substitution (DS) using the same volume of acetylation reagent. Regarding the thermoplastic properties, there is a clear correlation between the degree of substitution and the improvement of thermal stability. The samples are more thermally stable at higher degree of substitution [74].

In their studies, Fundador et al. proceed to acetylate hardwood xylan using DMAc/LiCl/pyridine system at different reaction times. They obtained a substitution degree of 1.6–2 and the improving of thermal stability and chloroform solubility of modified xylan. In addition, xylan acetate with a DS = 2.0 is able to be electrospun into nanofibres [75].

Eguesa et al. acetylated corn cob xylan using pyridine/acetic acid anhydride system. The acetylated corn cob xylan had about 1.9 DS and was used to prepare films by casting method. Obtained films had the improved water barrier and mechanical properties comparing with native corn cob xylan. Thus, the acetylation process allowed obtaining the superior values of tensile strength, 67 MPa, 13.4% of elongation at break, 2241 MPa of Young’s modulus and about 80–82° contact angle [76].

The acetylation of xylan in homogenous conditions was carried out by Belmokaddem et al., using dimethylformamide/LiCl/acetic anhydride system catalysed by dimethylaminopyridine. Depending on the reaction conditions, the degree of substitution was about 0.9–2.0. Due to the toxicity of dimethylformamide, this method is limited to be used at large scale. On the other hand, the heterogeneous conditions are environmentally friendly and offer an efficient way to prepare xylan acetate. In this case, the esterification was acidic catalysed using methanesulfonic acid and succinic anhydride when a maximum degree of substitution is obtained. The esterification reaction catalysed by methanesulfonic acid was also performed using propionic and hexanoic anhydrides when hydrophobic xylan acetate was obtained at average substitution degree [57].

In their research, Kaur et al. synthetized acetylated xylan extracted from rice straw by xylan reaction with acetic anhydride in the presence of pyridine. The reaction resulted in hydrophobic xylan with antioxidant properties and better film formation capacity [77].

Cheng et al. obtained xylan esters through acetylation with acetic anhydride, using as catalysts metal halides (AlCl_3_, FeCl_3_, SbCl_3_, SnCl_2_, ZnCl_2_) under different reaction conditions (Figure 1). The degree of substitution depends on the level of reaction temperature and the type of metal ion. Compared with iodine catalysts, the reactivity trend of metal halides is:

AlCl_3_ > FeCl_3_ > iodine > ZnCl_2_ > SbCl_3_ > SnCl_2_

**Scheme 1 polymers-15-02088-sch001:**
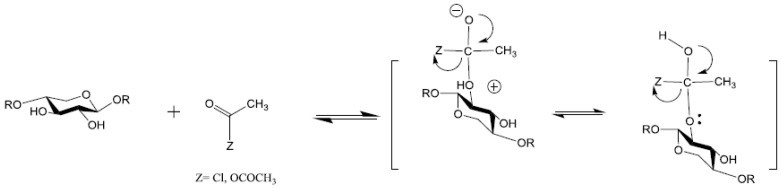
Possible mechanism for the metal chloride-catalysed acetylation of xylan [69].

For example, for AlCl_3_ and FeCl_3_, a degree of substitution of about two could be accomplished at a reaction temperature of 100 °C. For the other Lewis acids, a higher temperature (110 °C) was needed to achieve higher degrees of substitution [78,79].

### 2.4. Acetylation of Xylan Hemicelluloses with Ionic Liquids (ILs)

As mentioned above, hemicellulose can be esterified efficiently under homogeneous conditions, obtaining highly substituted derivatives with a satisfactory yield. However, the toxicity and high cost of organic solvents, such as dimethylformamide (DMF), dimethylsulfoxide (DMSO) or pyridine, limit their large-scale application. Therefore, due to the environmental pressure to obtain new sustainable products, a green and recyclable reaction system is preferable for the chemical modification of hemicellulose [80,81].

In this context, the use of ionic liquids (ILs) as solvent for polysaccharides and their enzymatic esterification are emerging fields that provide new opportunities and challenges for the modification of cellulose and hemicelluloses.

Ionic liquids (ILs) are green solvents or catalysts that are used to prepare biomaterials and generate bioenergy from lignocellulosic biomass. They have received significant attention in the field of green chemistry owing to their attractive properties, including their chemical and thermal stability, strong polarity, non-volatility and non-oxidizing nature [82].

Ionic liquids are water free organic salts with melting points of below 100 °C, being constituted completely of ions (ion-pairs with anions and cations). Due to of their low vapour pressure, low flammability, high viscosity and high thermal and chemical stability, along with the possibility of them being recovered and recycled, ionic liquids are considered of great significance for the development of environmentally friendly routes for the modification of hemicellulose [83,84].

According to the literature results, ionic liquids combined with ammonium, pyridinium and imidazolium cations are able to dissolve cellulose (Table 3) [85,86,87,88].

Although there is extensive research on the utilisation of ionic liquids for cellulose modification, only a few address to hemicelluloses; however, this topic has been gaining increasing interest in the last few years [90,91]. Through the utilisation of ionic liquids for hemicellulose modifications with or without catalysts, the high efficiency of the reaction can be achieved. Furthermore, ILs offer a potentially clean method for carrying out chemical reactions or processes [92,93].

The first use of ILs as an environmentally friendly solvent for the chemical modification of hemicelluloses was carried out by Ren et al., who acetylated wheat straw hemicelluloses with acetic anhydride using iodine as catalyst in ionic liquid based on 1-butyl-3-methylimidazolium chloride ([C4mim] Cl). In homogenous conditions and under different reaction temperatures, durations, and dosages of the catalyst and acetic anhydride, they obtained acetylated hemicelluloses with degree of substitution between 0.49 and 1.53, respectively, and a yield of 70.5% and 90.8%. The highest DS was obtained at a 20:1 reactant molar ratio at 100 °C, for 30 min and with 15% iodine. In these conditions, about 83% hydroxyl groups in native hemicelluloses were esterified. Through chemical modification, the thermal stability of the acetylated hemicelluloses was improved. It was also found that wheat straw hemicelluloses are completely soluble in ionic liquid, namely 1-butyl-3-methylimidazolium chloride, at up to a 2.6% (*w*/*w*) concentration at 90 °C and after 1.5 h [94].

In their studies, Ayoub et al. obtained acetylated hemicelluloses through acetylation using acetic anhydride in 1-allyl-3-methylimidazolium chloride ([Amim]^+^ Cl^−^) ionic liquid. They applied a complete homogeneous system and succeeded for the first time without a catalyst. The results emphasized increasing the DS and yield by increasing the reaction temperature from 30 °C to 80 °C and the duration from 1 h to 20 h. As a result, the increasing of the substitution degree improved the contact angle from 39° to 83°, as well as the thermal stability of the acetylated hemicelluloses. The obtained acetylated hemicellulose is soluble only in aprotic solvents, such as chloroform or dimethyl sulfoxide (DMSO) [95].

Zhu et al. applied a facile and novel method for hemicellulose acetylation by using mild acetylation in 1-ethyl-3-methylimidazolium acetate [Emim] OAc ionic liquid and dichloroacetyl chloride (Cl_2_AcCl) system. This is a one-pot method in which hemicelluloses react with mixed anhydride produced with EmimAc and Cl_2_AcCl rather than Cl_2_AcCl only. They obtained a fully water-soluble modified product with a degree of substitution value between 0.17 and 0.37. These acetylated hemicelluloses had increased viscosity, which indicated the potential for their utilisation in the coating or painting industry, as well as for chemical conversion [96].

The trans esterification with vinyl laurate conducted in [Emim] OAc ionic liquid was applied by Zhang et al. for the chemical modification of xylan hemicelluloses. The acetylated xylan hemicellulose with a degree of substitution about 1.85 was obtained at 80 °C after 2 h of chemical reaction. Using a solvent casting method, they prepared films with excellent hydrophobicity (contact angle about 120°) and enhanced mechanical properties (33.94 MPa of tensile strength and about 22.41% elongation at breaks). The xylan acetate hemicellulose films exhibited acceptable antioxidant activity and values of water vapour permeability (WVP) and oxygen permeability (OP), ranging from 1.59 ± 0.07 to 2.23 ± 0.11 (10^−10^·g/m·s·Pa) and 1.21 ± 0.04 to 4.24 ± 0.30 (cm^3^·μm/m^2^·d·kPa), respectively. Based on these properties, the obtained films were used for the packaging of green chilies. To evaluate the packaging efficiency of obtained films, the changes in weight, colour and texture were investigated. The results reveal that the shelf-life of green chilies was extended to 15 days after packing, indicating the promising applications of acetylated xylan as active bioplastic films for the shelf-life extension of fresh food products [97].

Using an optimized procedure, Stepan et al. achieved the acetylation of rye arabinoxylan and spruce arabinoglucuronoxylan in two systems of ionic liquids (ILs) in only a few minutes. The first system involved the direct dissolution of xylan into 1-ethyl-3-methylimidazolium dimethylphosphate ([Emim][Me_2_PO_4_]), followed by the addition of acetyl chloride/pyridine (AcCl/Pyr) and additional chloroform (CHCl_3_) as a co-solvent, when a degree of substitution of about 1.96 was obtained. The second system involved direct dissolution into the novel protic ionic liquid 1,5-diazabicyclo[4.3.0] non-5-enium acetate ([DBNH][OAc]), followed by acetic anhydride/1,5-diazabicyclo[4.3.0] non-5-ene (Ac_2_O/DBN), and no co-solvent was added. In this case, a complete acetylation was obtained with a value of substitution degree of about two [52].

The homogeneous acetylation reaction of hemicelluloses extracted from soya residues was performed by Chen et al., using for the first time a system with acetic anhydride in imidazolium-based ionic liquids. They used 1-allyl-3-methylimidazolium chloride (AMIMCl) and 1-butyl-3-methylimidazolium chloride (BMIMCl) and conducted the acetylation tests in different reaction conditions. They found the optimum reaction with a molar ratio of acetic anhydride to hydroxyl functionality in hemicelluloses of 14:1, a duration of 20 min and a temperature of 100 °C. In these conditions, the acetylation of hemicelluloses was satisfactory, with degree of substitution being about 1.68 and the reaction yield about 93.8%, respectively. In addition, the ionic liquids were able to be effectively reused several times. The obtained results emphasized that in comparison with BMIMCl, AMIMCl was the better reaction medium for obtaining acetylated hemicelluloses with relatively high DS. The cation of [AMIM]^+^ has a smaller ion size and a double bond, and it is commonly accepted that small and strong polar cation in ionic liquid preferably attack the oxygen atom of hemicellulose hydroxyl. In addition, the ionic liquid AMIMCl was more easily recovered after the removal of solvent (water) by evaporation and reused in a new acetylation reaction. This causes this type of ionic liquid to be potentially attractive for industrial applications [98,99]. The analysis of SEM images of native and acetylated hemicelluloses show that compared with the native hemicelluloses, which have a fibrous form, the acetylated hemicelluloses have a different morphology. Thus, as result of the homogeneous reaction, the acetylated hemicelluloses have a small, broken, irregular and smooth-faced structure. This indicated that native hemicelluloses were dissolved, and it was assumed that a smooth surface resulted from the acetyl groups (Figure 4).

The esterification of xylan-rich hemicelluloses in a homogenous system with maleic anhydride and 1-butyl-3-methylimidazolium chloride ([BMIM] Cl) ionic liquid using LiOH as catalyst was performed by Peng and co-workers [100]. Under different reaction times, temperatures, catalyst dosages and the molar ratios of maleic anhydride to anhydroxylose units in the hemicellulose, the DS of the modified hemicelluloses was between 0.095 and 0.75. Results obtained from FT-IR and C NMR spectroscopies confirmed the structure of hemicellulose derivatives with a carbon–carbon double bond and carboxyl groups. This can thus be considered an efficient method to prepare a novel and important functional biopolymer for environmentally friendly applications [100].

In their research, Wang et al. synthetized lauroylated hemicelluloses using lauroyl chloride in 1-butyl-3-methylimidazolium chloride ([BMIM] Cl) ionic liquid. By varying the reaction parameters, such as the molar ratio of lauroyl chloride to anhydroxylose units in hemicelluloses (0.5:1–3:1), the temperature of the reaction (80–90 °C) and the reaction time (15–90 min), they obtained modified hemicelluloses with a degree of substitution between 0.43 and 1.82. The process was applied to a homogeneous system and highly substituted hemicelluloses esters were obtained. It was found that during the esterification process, a significant degradation of hemicelluloses occurred, as the molar ratio, time and temperature were higher. After chemical modification, the morphological properties of hemicellulose were significantly improved and the thermal stability of obtained esters was lower, as compared with native hemicelluloses [101].

Zhang et al. investigated the butyrylation of hemicelluloses in homogenous system with butyryl chloride in 1-butyl-3-methylimidazolium chloride ([BMIM] Cl) ionic liquid using triethylamine as a neutralizer. By changing the reaction conditions, they obtained a degree of substitution between 0.91 and 1.89. The highest value of DS (1.89) was obtained at a 2:1 molar ratio of butyryl chloride to hydroxyl groups, at 90 °C and 120 min. The hydrophobicity and thermal stability of hemicellulose derivatives are improved with increasing degrees of substitution. This could thus be a beneficial alternative for the production of functional biopolymers [102].

A hemicellulose derivative was homogeneously synthesized by Yang et al. using cinnamic anhydride in 1-allyl-3-methylimidazolium chloride (AMIMCl) ionic liquid and different catalysts, such as NaOH, KOH and LiOH. The modified xylan derivatives with degrees of substitution between 0.11 and 0.57 were obtained by changing the reaction parameters. It was found that the optimum conditions were a molar ratio cinnamic anhydride/anhydroxylose of 6:1, a reaction temperature of 80 °C, a duration of 80 min and a level of catalyst of about 0.025 g NaOH. Compared with native xylan, the thermal stability of the xylan derivatives was reduced, and this could extend the application area of xylan hemicelluloses in wet-end papermaking, organic–inorganic composite films or for hydrogels synthesis [103].

The utilisation of phthalic anhydride in 1-allyl-3-methylidazium chloride [AmimCl] was investigated by Wang et al. for the esterification of bagasse hemicelluloses in a homogeneous system. The degree of substitution of hemicellulose samples ranged from 16.37% to 52.14%. It was observed that the side chains of hemicellulose were more easily esterified than the main chains [104]. The hydroxyl groups on glucuronic acids were more difficult to substitute than those on neutral sugars. The reactivity of the hydroxyl groups from C-2 and C-3 positions in anhydroxylose units was almost similar [105].

## 3. Enzymatic Modification of Hemicelluloses

Another green alternative for improving the physical strength and hydrophobicity of hemicellulose films is the enzymatic treatment [106]. Through enzymatic degradation, undesired branches from the hemicellulose backbone are removed and, as a result, the intermolecular aggregation of xylan chains is increased and the water solubility of xylan is reduced [65,107,108].

Due to the high complexity of the hemicelluloses structure, a variety of enzyme activities are needed for their complete breakdown. The most studied hemicellulose-active enzymes are xylan- and glucomannan-specific enzymes [109]. In many studies, it was reported that arabinase, laccase, β-galactosidase, lipases and cutinase were the main types of enzymes that were used for hemicellulose modification [110]. In other studies, the enzymes that modify xylan hemicelluloses were grouped in two classes: those enzymes that cleave the main xylan backbone, including endo-xylanase, exoxylanase and β-xylosidase, and those enzymes that break the side groups on the main xylan chain, called accessory enzymes. The accessory enzymes include α-L-arabinofuranosidase, α-Dglucuronidase, acetyl xylan esterase, ferulic acid esterase and ρ-coumaric acid esterase [111,112].

The accessory enzymes α-L-arabinofuranosidase and α-D-glucuronidase have been shown to modify the solubility of xylan precipitation into micro- and nanohydrogels by hydrolysing the respective side groups and modifying the side chain distribution pattern [113,114].

The morphological analysis of the modified xylan emphasized that through the enzymatic treatment, the gelling and plasticising properties are improved. As result of the removal of 4-O-methyl glucuronic acid from glucuronoxylans via enzymatic treatments, the viscosity is increased and the formation of hydrogels occurs. Arabinose and 4-O-methyl glucuronic acid removal from arabinoglucuronoxylan also resulted in the formation of insoluble xylan particles that are gravitationally settled. Based on these results, enzymatic modification could be considered to be mild and selective process that adds functionality to xylan hemicelluloses by reducing their water solubility and extending their utilisation for paper coatings, packaging films or hydrogel encapsulation matrices [115].

Hoije et al. performed the enzymatic modification of rye arabinoxylan using α-L-arabinofuranosidase. Through enzymatic treatment, the degree of polymerization of the arabinoxylans was not affected but the film formation ability was improved. Thus, the enzyme-treated arabinoxylans formed uniform films with crystalline structures, without the addition of external plasticizers. The reduction in the oxygen permeability of the enzymatic-treated xylan films and the obtaining of higher mechanical properties are promising features that could be used for packaging applications [116].

Stepan et al. reported for the first time the surface esterification of rye arabinoxylan films using enzymatic treatments. They investigated the changes in material properties by substituting the surface of arabinoxylan films into a homogenous system with a short acetyl chain and a long stearate chain to give a more expressed hydrophobicity to the surfaces. They also studied the activity of different lipases and cutinase enzymes on different alkyl chain lengths in the esterification reactions of hemicelluloses. When vinyl stearate was used as an acylation agent, the cutinase presented a higher activity against the vinyl acetate as a substrate and modified the surface of xylan films. The lipases were only capable of performing the surface stearation of the films. As a result of enzymatic modification, the contact angle of films exhibited increased initial hydrophobicity on their surfaces [117].

The modification of xyloglucan via the enzymatic removal of side-chain galactose residues was performed by Kochumalayil et al. As result of galactose removal, the properties of films made from modified xyloglucan, in terms of tensile, oxygen transmission rate and thermo-mechanical behaviour, were improved. For example, the oxygen permeability of samples with modified xylan was more than 80% lower compared with that of native xylan. Due to decreased solubility through enzymatic modification, modified xylan absorbs less humidity, which is beneficial for these derivatives when are used as food packaging materials [118].

Oinonen et al. reported a novel technique for increasing the molecular weight of different types of wood hemicelluloses, which creates new possibilities for utilizing this raw material source. The technique consists of the enzyme-catalysed cross-linking of aromatic moieties bound to the hemicelluloses. The cross-linking treatment resulted in significantly improving the mechanical and barrier properties of films made from modified spruce galactoglucomannan [119].

To improve its applicability and versatility, the molecular structure of xylan was modified by Bueno et al., using the action of accessory enzymes, such as α-glucuronidase and α-L-arabinofuranosidase. These enzymes hydrolyse pending groups in the xylan chain, decreasing their solubility and improving hydrophobic properties [120].

The enzymatic modification of xylan hemicelluloses was performed by Gama Gomes et al., who used two types of enzymes, namely α-L-arabinofuranosidase and α-D-glucuronidase, respectively, at low and high enzyme substrate conditions. The obtained functionalised xylans, presenting low solubility and changes in viscosity, as result of the release of glucuronic acids during enzymatic treatment [114].

In her thesis, Khan studied the efficiency of commercial enzymes mixtures, Celluclast and Pulpzyme HC in birch wood xylan hydrolysis using different reaction parameters. The Celluclast enzyme displays mainly β-xylosidase activity and Pulpzyme has β-1–4 xylanase activity. Based on the obtained results, he concluded that the most important factor is pH and obtained the highest yield at pH 5. The temperature and ratio of enzymes are also important parameters [121].

The enzymatic hydrolysis of soluble arabinoxylan into insoluble hydrogels was achieved by Thokozani, who used the α-L-arabinofuranosidase and α-D-glucuronidase enzymatic systems. These hydrogels can function as delivery systems and matrices for chemical or bioactive substances. The biological formation of hemicellulose entrapment matrices using side-chain-removing enzymes offers a green alternative that allows for the protection of sensitive encapsulated substances, such as bioactive agents [111,122].

Heikkinen et al. utilised enzymes for partially removing arabinose units from wheat arabinoxylan. As result, the water solubility of modified xylan was reduced and mechanical, thermal and barrier properties of the resultant hemicellulose-based films were improved [123].

Pitkänen et al. modified the structure of arabinoxylans from wheat and rye endosperms using specific enzymatic treatments with α-L-arabinofuranosidase. As result, the water solubility of xylan was reduced [124,125].

## 4. Conclusions

This paper presents an overview of the process of the chemical modification of xylan hemicellulose, with a focus on more environmentally friendly acetylation methods in order to improve its functional properties, as required by food packaging applications. Therefore, the chemical modification of xylan hemicelluloses by acetylation with anhydrides, ionic liquids and enzymatic treatments was described in depth.

The literature survey reveals that the thermal stability, water and gas barrier properties, as well as mechanical resistance and flexibility, of films from hemicellulose derivatives are improved through acetylation as result of the replacement of hydroxyl groups with hydrophobic esters groups.

The interest in the utilisation of ionic liquids for hemicellulose modification has increased rapidly in recent years in view of obtaining of derivatives with the maximum degree of substitution and improved properties within only a few minutes of a reaction. In addition, the possibility of recovering and recycling hemicellulose is an important reason for a method to be considered an environmentally friendly. Based on reported studies, ionic liquids with ammonium, pyridinium and imidazolium cations are able to dissolve polysaccharides (i.e., cellulose). Regarding hemicellulose dissolution, appropriate results were obtained through the utilisation of imidazolium salts.

Through enzymatic modification, the water solubility of xylan hemicellulose is reduced, as result of the removal of undesired branches from the hemicellulose backbone, which increases the intermolecular aggregation of xylan chains. The most studied active enzymes for hemicellulose modification are xylan- and glucomannan-specific enzymes, such as α-L-arabinofuranosidase and α-D-glucuronidase.

In terms of environmental impact, hemicellulose derivatives could be a sustainable alternative to existing synthetic polymers and extend the application areas of hemicelluloses as biopolymers in paper coatings and biodegradable films for the protection of food products.

## Data Availability

All data are provided in the paper.

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
