# Peer review of "Green Approaches on Modification of Xylan Hemicellulose to Enhance the Functional Properties for Food Packaging Materials—A Review"

_polymers, 2023, doi:10.3390/polym15092088_

Round 1
Reviewer 1 Report
the author try to discuss about the Green approaches on acetylation of xylan hemicellulose to en-2 hance the functional properties for food packaging materials. the manuscript needs revision. the comments are
the introduction is too long.
it will help to understand if author could include a comparison table.
review article should focus on recent work done. there are many old references , author should keep it mind.
author should include more images for each section.
there is a lack of information on food packaging.
scale bar is not clear in fig 4.
Author Response
Dear reviewer,
Thank you very much for your useful observations, comments and recommendations. We deeply appreciate your evaluation.Your fruitful comments helped us to improve the overall quality of the manuscript.
Please, find bellow, point by point, the response at your comments:
Reviewer comments
- the introduction is too long.
- it will help to understand if author could include a comparison table.
- review article should focus on recent work done. there are many old references , author should keep it mind.
- author should include more images for each section.
- there is a lack of information on food packaging.
- scale bar is not clear in fig 4.
Response
Whole manuscript was depply checked and revised.
- The introduction was shorted according with your suggestion. All the modifications are highlighted in the revised manuscript with red color.
- In the revised manuscript two tables were introduced in the Section 2.3. which present the comparative properties of acetylated xylan versus native xylan and PLA or cellulose acetate. (please, see table 1 and table 2).
- All citations were revised and 19 of recent references were introduced. (2020-2023)
- In the revised manuscript the information regarding the food packaging were highlighted with red color in each section. Generally, the modification of hemicelluloses has as result the improving of barrier properties and hydrophobicity of xylan hemicellulose, features required for food packaging .
- The figure 4 was changed with visible scale bar.
All the modifications are highlighted in the revised manuscript with red color.

Reviewer 2 Report
Review “Green approaches on acetylation of xylan hemicellulose to enhance the functional properties for food packaging materials. A review” corresponds to the Polymers edition in terms of content: it describes how the functionalization of xylan hemicellulose for the subsequent specific application of “food packaging materials”. The positive side of the review is its timeliness, since, as a rule, they analyze methods for obtaining packaging materials from cellulose, ignoring the second biopolymer from lignocellulosic biomass, xylan hemicellulose.
This article presents an analysis of publications in the amount of 106 pieces, summarizing the methods of acetylation in various media and enzyme modification, but only 18 publications date from 2020-2023. The reviewer has a question: for what reason are numerous sources for 2020-2023, presented in Figure 2 in the amount of about 250 pieces, not cited in the review? It follows from the caption that they are all devoted to "hemicelluloses chemical modification". Thus, this review can be published after the authors follow the recommendations below.
Recommendations:
1. The review includes, in addition to acetylation, one more method of modification, so the name must be changed. The least traumatic method is the replacement of one word, that is, “Green approaches on MODIFICATION of xylan hemicellulose to enhance the functional properties for food packaging materials. A review"
2. The abstract should be rewritten emphasizing the scientific value of the review. "Green" statements contribute to the relevance of the topic under discussion, and scientific novelty is not expressed.
3. The text should be supplemented with a small section with information about the availability of xylan hemicellulose on an industrial scale.
4. The list of sources should be expanded to include recent publications, especially since Figure 2 indicates a sufficient number of them.
Author Response
Dear reviewer,
Thank you very much for your useful observations, comments and recommendations. We deeply appreciate your evaluation.Your fruitful comments helped us to improve the overall quality of the manuscript.
Please, find bellow, point by point, the response at your comments:
Reviewer comments
- The review includes, in addition to acetylation, one more method of modification, so the name must be changed. The least traumatic method is the replacement of one word, that is, “Green approaches on MODIFICATION of xylan hemicellulose to enhance the functional properties for food packaging materials. A review"
The abstract should be rewritten emphasizing the scientific value of the review. "Green" statements contribute to the relevance of the topic under discussion, and scientific novelty is not expressed.
The text should be supplemented with a small section with information about the availability of xylan hemicellulose on an industrial scale.
The list of sources should be expanded to include recent publications, especially since Figure 2 indicates a sufficient number of them.
Response
Whole manuscript was depply checked and revised.
- The title of manuscript was modified according your suggestion, and the section 2 was restructured according to new title.
- The abstract was modified to emphasize the scientific value of the manuscript.
- In the Section 2 was introduced a new subsection 1. Xylan hemicellulose – potential applications and industrial availability where the information about the commercial availability of xylan hemicelluloses were introduced.
- All citations were revised and 19 of recent references were introduced. (2020-2023) It is true that the number of publications presented in the Figure 2 is large, but this contains all the publication regarding on the hemicellulose modification. To be more specific, in the legend of Figure 2 it was mentioned the number of publications addressed only to acetylation of hemicelluloses and ionic liquids acetylation.
All the modifications are highlighted in the revised manuscript with red color.

Round 2
Reviewer 1 Report
authors have addressed all the concerns.